# Modalities Differentiation of Pain Perception Following Ischemic Stroke: Decreased Pressure Pain Perception

**DOI:** 10.3390/biomedicines13092241

**Published:** 2025-09-11

**Authors:** Yongkang Zhi, Chen Zhao, Yu Zhang, Jianzhang Ni, Ming Zhang, Dongsheng Fan, Yazhuo Kong

**Affiliations:** 1CAS Key Laboratory of Behavioral Science, Institute of Psychology, Chinese Academy of Sciences, Beijing 100101, China; zhiyk@psych.ac.cn; 2Department of Psychology, University of Chinese Academy of Sciences, Beijing 100049, China; 3Peking University Third Hospital, Beijing 100091, China; zhaochen0121@163.com; 4Department of Applied Psychology, School of Humanities and Social Sciences, Beijing Forestry University, Beijing 100083, China; zhangyu2023psy@bjfu.edu.cn; 5Department of Psychiatry, Chinese University of Hong Kong, Hong Kong, China; j.ni@link.cuhk.edu.hk; 6Wellcome Centre for Integrative Neuroimaging, FMRIB, Nuffield Department of Clinical Neurosciences, University of Oxford, Oxford OX3 9DU, UK

**Keywords:** ischemic stroke, hypoalgesia, nociception, pain thresholds, pain integration

## Abstract

**Background/Objectives**: Ischemic stroke frequently leads to somatosensory impairments and abnormal pain perception. Meanwhile, pain perception can be evoked through multiple somatosensory modalities, each mediated by distinct neural pathways. Despite this understanding, current research investigating stroke-induced alterations in pain perception across different modalities of noxious stimulation remains insufficient, particularly concerning responses to varying stimulus intensities (including both sub-threshold and supra-threshold levels). **Methods**: In this study (March 2023 to July 2024), we enrolled 30 ischemic stroke patients and 35 matched controls and employed two modalities of noxious stimuli (e.g., heat stimuli were delivered using the Medoc CHEPS and pressure stimuli were administered via an MRI-Compatible Foot-Sole Stimulator) to systematically evaluate post-stroke changes in pain perception through two experiments. We compared self-reported pain sensitivity, somatosensory thresholds (i.e., warmth and pressure), and pain thresholds (i.e., heat and pressure pain) between ischemic stroke patients and healthy controls in Experiment 1. We focused on pain perception when participants simultaneously experienced heat and pressure in Experiment 2. **Results**: Experiment 1 showed an absence of a significant correlation between heat and pressure pain thresholds in stroke patients, but this correlation could be observed in healthy controls. Notably, stroke patients had an impairment in pain perception of pressure stimulation at supra-threshold intensities. Experiment 2 observed a similar facilitative pain integration in patients as healthy controls when they perceived heat and pressure stimuli jointly and simultaneously. **Conclusions**: These findings provide valuable insights into pain perception following a stroke, highlighting the need for tailored evaluation strategies considering the differences in somatosensory modality damage.

## 1. Introduction

Stroke represents a significant global health challenge, often leading to disability and an increased risk of recurrence [1,2]. Ischemic stroke in particular, is the leading cause of death worldwide and is associated with complex, chronic disabilities [3,4]. Stroke patients frequently experience somatosensory deficits, which may manifest as either heightened pain sensitivity or, conversely, a marked reduction or complete loss of nociception [5,6,7], which can profoundly affect their quality of life [8,9,10]. Comprehensive somatosensory evaluation post-stroke is critical for early detection and prevention of potential complications arising from noxious stimuli in patients with sensorimotor impairments [11,12].

Multimodal somatosensory processing involves various types of nerve fibers [13]. Small fibers, such as C and Aδ, primarily respond to thermal, mechanical, and chemical stimuli [14,15], while large fibers, like Aβ, are associated with sensations of touch and pressure on the skin [16,17]. These different somatosensory modalities are vital for the daily functioning of stroke patients [18]. Many stroke survivors experience somatosensory deficits [19]; therefore, it is essential to clarify modality-specific assessments of painful peripheral stimulation and how these relate to somatosensory profiles and post-stroke complications.

As the intensity of somatosensory stimuli—such as heat, cold, pressure, and chemicals—increases, they can provoke pain [20], serving as an important warning system in potentially dangerous situations [21]. Somatosensory impairments in stroke survivors may lead to contradictory pain experiences, heightened pain sensitivity, or diminished pain perception [6,22,23]. Research indicates that pain perception in response to noxious stimuli can vary based on the type of stimulation [24]; however, this aspect has received insufficient attention in stroke patients [25]. Additionally, there is a scarcity of studies examining the perception of noxious stimuli at sub- or supra-threshold levels, which could shed light on the perceptual changes associated with different somatosensory modalities. Addressing these gaps is crucial, as it may provide valuable clinical insights for developing targeted treatments during stroke rehabilitation.

The perception of objects relies on integrating multiple sensory modalities [25]. Our nervous system combines information from various sensory inputs to form a unified understanding of our environment, allowing for quick and coordinated responses [26,27]. Somatosensory experiences—such as pain, temperature, and touch—are activated by specific stimuli [20,28]. Moreover, different somatosensory modalities can interact during perception [29]. However, it is unclear whether this integration remains intact in stroke patients when they are exposed to multiple noxious stimuli.

Given the potential discrepancies in multimodal somatosensory sensations following a stroke, the current study aimed to investigate changes in pain perception associated with dual-modal somatosensory stimulation (i.e., heat and pressure) based on small and large fibers with various intensities through two experiments. In Experiment 1, we compared self-reported pain sensitivity, somatosensory thresholds (i.e., warmth and pressure), and pain thresholds (i.e., heat and pressure pain) between ischemic stroke patients and healthy controls. Experiment 2 focused on pain perception when participants simultaneously experienced heat and pressure, allowing us to assess the ability of stroke patients to integrate pain at a higher cognitive level. We expected that stroke patients and healthy controls would differ in their perception of noxious stimuli of different modalities at the perception level, but not in their integrated pain at the cognitive level.

## 2. Materials and Methods

### 2.1. Participants

A priori power analysis was conducted using G*Power 3.1 to determine appropriate sample sizes for both experiments, based on the repeated-measures ANOVA guidelines proposed by Bartlett (2022; “https://osf.io/zqphw/ (accessed on 22 January 2023)”) [30]. For Experiment 1, which employed a mixed design with one between-participants factor (group) and two within-participant factors (stimulation modality and intensity), a sample size of 12 participants per group was estimated to provide 80% power to detect a medium effect size (*f* = 0.25) at *α* = 0.05. Similarly, for Experiment 2’s design featuring one between-participants factor (group) and one within-participants factor (condition), a total sample size of 28 participants was calculated to achieve equivalent statistical power (80%) for detecting medium effects (*f* = 0.25) at the same significance threshold. We enrolled 30 patients (28 males, 2 females) with ischemic stroke from the Neurology Department of Peking University Third Hospital between March 2023 and July 2024 (see Table 1 and Appendix A). Additionally, 35 demographically matched healthy controls (33 males, 2 females) were recruited. All 35 healthy controls and 30 stroke patients participated in Experiment 1 and Experiment 2. The study was conducted in accordance with the Declaration of Helsinki, and the protocol was approved by the ethics committees of the Institute of Psychology at the Chinese Academy of Sciences (H22139) and Peking University Sixth Hospital (2023-084-01). Informed consent was obtained from all participants involved in the study.

Inclusion criteria for stroke patients were as follows: (1) a confirmed diagnosis of ischemic stroke with a visible unilateral lesion on MRI or CT; (2) absence of severe neurological disorders (e.g., brain tumor, trauma, multiple sclerosis), physical lesions, or psychiatric conditions (e.g., schizophrenia); (3) no consciousness, memory, or language disorders; (4) National Institutes of Health Stroke Scale (NIHSS) scores ≤ 5 and Modified Rankin Scale (MRS) scores ≤ 2 [31,32]; (5) stroke occurrence within 2 months prior; (6) aged 18–75; (7) no external injuries, frostbite, or sores on the tops or soles of the feet; (8) no spontaneous pain; and (9) informed consent signed by the patient. Exclusion criteria included: (1) unconscious or intubated patients; (2) those with dementia, cognitive impairment, or aphasia who cannot identify their stroke symptoms; (3) patients who received electroconvulsive therapy (ECT) in the last six months; (4) individuals taking medications that may affect pain perception (e.g., glucocorticoids, β-blockers, opioids, central stimulants); and (5) those unable to read or understand the informed consent form. We recorded patients’ medication and ensured they had taken it more than 4 h before participating in the experiments. Healthy controls were selected based on the absence of a history of stroke, severe primary diseases (e.g., cardiovascular issues, syncope, skin diseases, seizures), any surgeries in the last six months, and no external injuries to the feet. All participants were instructed to refrain from alcohol and pain medications for at least 24 h prior to the experiment.

### 2.2. Experiment 1

**Heat stimulation.** All heat pain stimuli were generated using the Medoc 9 cm^2^ Contact Heat-Evoked Potential Stimulator (CHEPS; Medoc Ltd., Ramat Yishai, Israel, 2019). To determine the heat pain threshold, participants’ right foot was tested five times, specifically 3 cm above the middle toes (see Figure 1A). The temperature was gradually increased at a rate of 0.5 °C/s, starting from 32 °C, until participants indicated the onset of pain by pressing a button to turn off the device. In the main experiment, participants experienced 45 heat pulses, each lasting 5 s, with temperature rising and falling at a rate of 40 °C/s. The heat stimuli were delivered at nine different intensities: 42 °C, 43 °C, 44 °C, 45 °C, 46 °C, 47 °C, 48 °C, 49 °C, and 50 °C [33]. Participants reported their pain levels for these brief heat stimuli using a numerical pain rating scale from 0 to 10 (0 = no feeling, 1 = a feeling of warmth, 2 = a feeling of heat, 3 = a feeling of hotness, 4 = just a feeling of pain, 10 = the worst pain imaginable) [34]. Values from 4 to 10 corresponded to progressively increasing pain levels. Due to individual tolerance variations, six patients and five healthy controls could not withstand the higher temperatures. Consequently, they were tested within a lower temperature range (39 °C to 47 °C), but pain ratings of the typical intensities (42–47 °C) were recorded and analyzed.

**Pressure stimulation.** All pressure stimuli were delivered using the MRI-Compatible Foot-Sole Stimulation System (custom-built, 2022) [35]. To assess the pressure pain threshold, five progressively increasing pressure stimuli were applied to the sole of the left foot, 3 cm below the middle toe, at a rate of 12 N/s, starting from 100 N. Participants indicated the onset of pain through oral reports. In the main experiment, participants received 45 pressure pulses, each lasting 5 s, at intensities of 100 N, 120 N, 140 N, 160 N, 170 N, 180 N, 210 N, 220 N, and 250 N [33]. Pain perception was evaluated using a numerical pain rating scale from 0 to 10 (0 = no feeling, 1 = a feeling of touching, 2 = a moderate feeling of pressure, 3 = a strong feeling of pressure, 4 = just a feeling of pain, 10 = the worst pain imaginable) [33].

**Procedure.** Participants first completed the Pain Sensitivity Scale (PSQ, e.g., “Imagine you burn your tongue on a very hot drink”, response from 1 = no pain to 10 = pain as bad as it could be [36], Cronbach’s α = 0.93). Following the pain threshold assessment, participants engaged in a two-stage main experiment (see Figure 1B). In Stage 1, we measured participants’ responses to one type of stimuli (e.g., heat stimulation) through 45 pulses presented in three blocks at nine different intensities, arranged in a pseudo-randomized order. Stimuli presentation and manual response tracking were managed using E-Prime 3.0 (Psychological Software Tools, Inc., Pittsburgh, PA, USA). Each trial began with a white fixation cross displayed for 1 s, followed by an instruction (e.g., “heat stimulation on”) for 5 s during which a heat pulse was applied to the right foot. Participants then rated their sensations using the numerical pain rating scale, which remained visible for 5 s. After this, a black background screen appeared 10 s before the next trial began. Stage 2 assessed participants’ responses to the other stimulation, following a similar procedure to Stage 1, but with the instruction changed to “pressure stimulation on”. The order of the two stimulus types was balanced across participants to ensure a comprehensive evaluation.

### 2.3. Experiment 2

Upon completion of Experiment 1, all patients and healthy controls participated in Experiment 2, except for six patients who required an interval due to fatigue. Specifically, one patient had a delay of 109 days, another had 7 days, and four patients had a delay of 1 day.

**Stimulation.** In Experiment 2, we employed similar heat and pressure stimuli as in Experiment 1. The intensities for the main experiment were first determined on participants’ feet. For heat stimulation, the temperature at which participants reported feeling moderate pain (i.e., a rating of 7) three times over was assessed through a 5 s heat application with a rise rate of 40 °C/s, targeting the right foot, 3 cm above the middle toe. For pressure stimulation, the intensity was defined in the same manner: participants indicated the point of moderate pain (also a rating of 7) during a 5 s pressure application on the sole of the left foot, 3 cm below the middle toe.

**Procedure.** We assessed the stimulation intensities separately on the right and left feet. The main experiment consisted of 45 trials, organized into three blocks and presented in a pseudo-randomized order. Participants experienced three conditions: (1) pain perception from heat stimulation alone (Heat pain), (2) pain perception from pressure stimulation alone (Pressure pain), and (3) comprehensive pain perception from simultaneous heat and pressure stimuli (Combined pain) (see Figure 1C). In each trial, a white fixation cross was displayed on the screen for 1 s, followed by a 5 s instruction indicating the type of stimulation. After this, a black background screen was shown for 5 s before a heat pulse was applied to the right foot for 5 s, either alone or in conjunction with a pressure pulse on the left foot, or a pressure pulse was delivered without the heat pulse. A black background screen then appeared for 10 s. Participants were subsequently prompted to rate their pain perception—considering the heat pulse, the pressure pulse, or the combined stimuli—using a numerical pain rating scale, which was displayed for 5 s. Following this rating, a black background screen appeared for another 10 s before the next trial began.

### 2.4. Statistical Analysis

Statistical analyses were conducted using SPSS 26.0 (IBM Corp., Armonk, NY, USA). Independent samples *t*-tests were employed to examine group differences in demographic characteristics, pain thresholds (heat and pressure), and stimulation intensities between stroke patients and healthy controls. Pearson correlation analyses assessed the relationship between heat and pressure pain thresholds within each group. For Experiment 1, between-group differences in pain ratings at each stimulus intensity were evaluated using independent samples *t*-tests. Experiment 2 utilized a two-way repeated measures ANOVA with between-subjects factor Group (stroke/control) and within-subjects factor Condition (heat pain/pressure pain/combined pain). Post hoc pairwise comparisons used Bonferroni adjustment. Significance was set at *p* < 0.05.

## 3. Results

### 3.1. Clinical and Neuroimaging Profiles of the Stroke Cohort

The cohort comprised 30 ischemic stroke patients (28 males, 2 females; mean age 55.67 ± 11.31 years, range 27–77) examined at a mean of 11.07 ± 8.13 days post-onset. All presented mild deficits (NIHSS 1.31 ± 1.47; MRS ≤ 2). Lesion topography revealed thalamic involvement in seven patients (23.3%), basal ganglia in seven (23.3%), frontal lobe in six (20.0%), cerebellum in three (10.0%), posterior limb of internal capsule (PLIC) in three (10.0%), and other sites (temporoparietal junction/pons/lacunar infarction) in four (13.3%). Hypertension (46.7%) and diabetes (16.7%) were predominant comorbidities. Full clinical and radiological profiles are provided in Appendix A.

### 3.2. Variability in Pain Threshold Correlations Across Stimulation Modalities

In Experiment 1, we compared the pain thresholds for heat and pressure stimulation between stroke patients and healthy controls. The independent samples *t*-test showed no significant differences in heat or pressure pain thresholds between the two groups (Table 1). However, a significant positive correlation was found in healthy controls, where the heat pain threshold was positively correlated with the pressure pain threshold (*r* = 0.44, *p* = 0.011, see Figure 2A). In contrast, this correlation was not significant in stroke patients (*r* = 0.20, *p* = 0.341). The different threshold correlations between the two groups suggest a shift in the relationship between pain perception modalities in stroke patients.

### 3.3. Decreased Pressure Pain Perception Following Stroke

To investigate whether stroke patients exhibit a decline in pain perception for dual-modal noxious stimuli, we compared both self-reported pain sensitivity and pain perception across various sub-threshold and supra-threshold pain intensities. The analysis revealed no significant difference in pain sensitivity (PSQ score) between stroke patients and healthy controls (Table 1). When calculating the mean pain ratings for heat and pressure stimuli at each intensity level, we found that pain perception increased with higher stimulation intensities (Figure 2B), consistent with established perception patterns [33]. There was no significant difference in the perception of heat stimulation between stroke patients and healthy controls, *p* ≥ 0.19.

However, compared to healthy controls, stroke patients provided lower pain ratings for pressure stimuli at supra-threshold intensities: *t*_(63)_ = −2.18, *p*_170N_ = 0.033, *d* = 0.55 for 170 N; *t*_(63)_ = −2.31, *p*_180N_ = 0.024, *d* = 0.57 for 180 N; *t*_(63)_ = −2.34, *p*_210N_ = 0.023, *d* = 0.58 for 210 N; *t*_(63)_ = −2.20, *p*_220N_ = 0.032, *d* = 0.54 for 220 N; *t*_(63)_ = −1.87, *p*_250N_ = 0.066 for 250 N, but not for those at sub-threshold intensities, e.g., *p*_160N_ ≥ 0.127 (see Table 2). These findings suggest a compromised perception of pressure pain but not pressure stimuli in stroke patients.

### 3.4. Unimpaired Pain Integration Following Stroke

In Experiment 2, a two-way repeated measures ANOVA was conducted, examining the effect of conditions (heat pain, pressure pain, and combined pain) as a within-participant variable and group (stroke, healthy) as a between-participant variable. The analysis revealed a significant main effect of condition on pain ratings, *F*_(2,126)_ = 13.70, *p* < 0.001, *η_p_^2^* = 0.18, but no significant main effect of group or interaction effect, *F*_(1,63)_ = 0.19, *p* = 0.664; *F*_(2,126)_ = 0.51, *p* = 0.604. Further analysis indicated that pain ratings in the combined condition (stroke: 5.12 ± 1.93 vs. health: 5.00 ± 1.58) were significantly higher than those for heat pain (stroke: 4.76 ± 1.72 vs. health: 4.62 ± 1.53), *t*_(29)*stroke*_ = 2.96, *p* = 0.013, *d* = 0.20; *t*_(34)*health*_ = 3.34, *p* = 0.004, *d* = 0.24, and pressure pain (stroke: 4.89 ± 1.74 vs. health: 4.62 ± 1.62), *t*_(29)*stroke*_ = 2.46, *p* = 0.049, *d* = 0.13; *t*_(34)*health*_ = 4.23, *p* < 0.001, *d* = 0.24, for both groups (Figure 2C). These findings demonstrate a pain integration phenomenon, indicating a facilitative effect on pain perception when both heat and pressure stimuli were experienced simultaneously by stroke patients and healthy controls.

## 4. Discussion

The current study investigated the differences in multimodal pain perception (i.e., pain sensitivity, heat pain threshold, pressure pain threshold, heat pain perception, pressure pain perception, and pain integration) in patients after ischemic stroke by examining dual-modal heat and pressure stimuli targeting small and large fibers. We found no significant differences in heat and pressure pain thresholds between stroke patients and healthy controls; however, a significant correlation between the two thresholds was observed in healthy controls but not in stroke patients, implying a potential change in the relationship between pain modalities in stroke patients. Additionally, stroke patients reported significantly lower pressure pain perception compared to healthy controls. Despite this, both groups exhibited a similar facilitative effect on pain perception when experiencing heat and pressure simultaneously, suggesting that the integrative function of multimodal somatosensory processing remains intact in stroke patients. These findings enhance our understanding of somatosensory disorders post-stroke and highlight the need for targeted rehabilitation strategies that consider discrepancies in pain perception across different modalities.

Unlike healthy controls, stroke patients showed no significant correlation between heat and pressure pain thresholds. Previous studies have demonstrated that pain thresholds across various modalities—such as heat, cold, capsaicin, pressure, and electrical stimulation—significantly correlate with healthy individuals [37,38,39]. For stroke patients, pain experiences can vary widely and are influenced by the disease process [23,40]. This altered relationship between pain thresholds suggests a potential dissociation in pain perception across different modalities following a stroke. Although the stroke itself may not directly impact the overall pain threshold, it appears to influence pain perception across modalities differentially. These findings indicate that even a stroke of seemingly mild severity can affect an individual’s pain perception.

Interestingly, stroke patients demonstrated significantly reduced pain perception in response to supra-threshold pressure stimuli, despite no abnormal changes in their pressure pain thresholds. This finding contrasts with previous research identifying light-touch deficits in stroke patients [9]. Multiple somatosensory effects induced by stimuli based on different fibers could be related to pain threshold variation [38]. Pain perception also involves central nervous system processing [41,42], unlike pain thresholds, which reflect peripheral nociceptive inputs. Given that stroke lesions typically occur within the central nervous system [27], they are likely to influence pain perception more than pain thresholds. Pain perception is fundamentally based on nociception, which assesses the characteristics of noxious stimuli, such as intensity [41]. However, the processes that shape perception can be modulated, resulting in either lower or higher perceptions than expected [41]—especially in cases of central nervous system damage from a stroke.

Consistent with previous research highlighting significant pressure pain sensory loss in stroke patients [6,23], our findings confirmed reduced pressure pain ratings in response to noxious stimuli. Heat and pressure stimuli engage different nociceptive pathways: heat primarily activates skin nociceptors, while pressure stimulates both skin and deep tissue nociceptors [22,42,43]. This distinction may be a physiological basis to explain the varied impact of stroke on these two pain modalities. Additionally, heat pain is inherently more complex than pressure pain, encompassing both first pain (sharp, mediated by Aδ fibers) and second pain (diffuse burning, mediated by C fibers) [44]. As such, the multifaceted nature of heat pain may attenuate the manifestations of stroke effects compared to the more straightforward perceptions of pressure pain. Consequently, the impact of stroke appears to be more pronounced in the context of pressure pain.

Pain processing systems are designed not only to discriminate nociceptive inputs but also to integrate and summarize this information [44]. In Experiment 2, we found that stroke patients demonstrated a similar capacity for pain integration as healthy controls. This integration of dual noxious inputs often results in an amplified pain experience [45]. Previous research has even indicated enhanced multisensory integration, such as in visual and auditory modalities [46,47], which may stem from deficits in top-down attentional control that allows for more cross-modal processing [46]. In this context, dividing attention across multiple sensory modalities is crucial for effective multisensory integration, whereas restricting attention to a single modality can enhance integration [48]. Interestingly, our results suggest that stroke patients in this study could effectively restrict their attention to single somatosensory stimuli, at least at the behavioral level. Although stroke is typically associated with alterations in sensation, perception, and cognition, as well as changes in brain anatomy and physiology [49,50], our study indicates that while there might be basic sensory degeneration, the cognitive-level functions related to integration have not yet been adversely affected. Integration primarily occurs within bilateral insular-somatosensory networks [51]. Crucially, multimodal integration engages supra-modal networks—notably the dorsolateral prefrontal cortex (DLPFC) and anterior cingulate cortex (ACC)—which assess stimulus salience and allocate attentional resources to concurrent inputs [52]. Our findings indicate that these networks may retain functional resilience, compensating for disruptions in primary sensory pathways. This aligns with evidence that cognitive-evaluative pain processing (e.g., contextual modulation) is less impaired post-stroke than sensory-discriminative components [53], likely due to redundant frontal-limbic connectivity [54].

The current study has several limitations. First, the variability in stroke lesion locations among participants means we did not account for the potential impact of lesion site on pain sensitivity. Future research should focus on recruiting patients with a specific lesion location or conducting subgroup analyses with larger sample sizes to address this gap. Second, there is an imbalance in the gender distribution of participants, with a predominance of male patients in our study. This limitation affects the generalizability of our findings, which are primarily based on male patients, although males have a higher incidence of stroke than females, females experience greater stroke severity and higher mortality than males [55]. Including a balanced representation of female patients in future studies would enhance our understanding of somatosensory profiles, particularly regarding the relative severity of impairment across different modalities. Third, heat and pressure stimuli were consistently applied to fixed positions on the right and left feet, respectively. Given the variability in stroke lesion locations and their potentially differential effects on ipsilateral versus contralateral somatosensory processing, this approach may have introduced side-specific confounding factors. Future studies should incorporate stimulus laterality as a controlled experimental variable, with stimulation sites carefully selected based on individual stroke topography to better elucidate hemispheric-specific pain processing alterations. Fourth, the patients in our study had relatively mild symptoms, and many pain dimensions showed no significant change; however, we did observe alterations in pressure pain perception, even at mild levels. This suggests that stroke may impact different pain modalities at varying rates, with pressure pain perception potentially being affected earlier than other types. Future research should include patients with moderate to severe strokes to investigate this phenomenon further. Fifth, while our study focused on ischemic stroke, hemorrhagic stroke may yield distinct pain profiles. Hemorrhagic stroke involves tissue compression and inflammation, potentially exacerbating central pain modulation. For example, thalamic hemorrhage can lead to the development of central post-stroke pain [56], whereas ischemic lesions in our cohort primarily affected pressure nociception. Future studies should compare sensory profiles across stroke subtypes.

In conclusion, this study highlights a dissociation in multimodal pain perception following stroke. Specifically, stroke patients exhibited impaired pressure pain perception, which relies on large fibers, compared to healthy controls. Clinically, these findings emphasize the importance of employing multimodal sensory testing (beyond standard clinical measures) during post-stroke assessment. Identifying a specific deficit in pressure pain perception could serve as an early biomarker for sensory dysfunction and guide individualized rehabilitation strategies. For instance, patients with such deficits might benefit from targeted sensory re-education protocols and early interventions aimed at preventing maladaptive neuroplasticity that can lead to chronic pain conditions. These findings suggest that targeted sensory training should be incorporated into the rehabilitation prescription for stroke patients, which may be beneficial in predicting chronic pain and preventing potential central or peripheral sensitization.

## Figures and Tables

**Figure 1 biomedicines-13-02241-f001:**
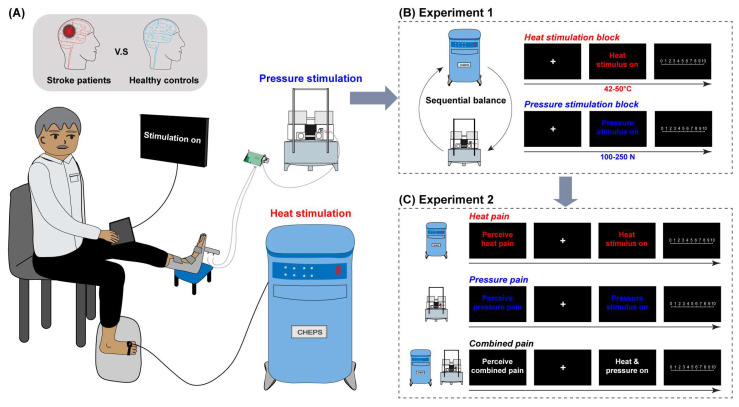
Experimental setup and conditions in the current study. (**A**) Illustration of the experimental apparatus used for delivering thermal and pressure stimuli. (**B**) Schematic of the experimental design and conditions in Experiment 1. (**C**) Schematic of the experimental design and conditions in Experiment 2 (The “+” symbol indicates fixation periods requiring participants to maintain attention).

**Figure 2 biomedicines-13-02241-f002:**
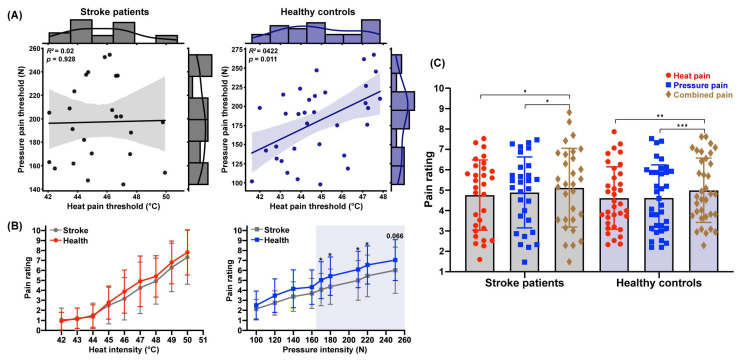
Comparison of pain perception between stroke patients and healthy controls. (**A**) Correlation between heat and pressure pain thresholds. (**B**) Pain ratings of heat and pressure stimuli at varying intensities. (**C**) Pain integration resulting from dual-modal noxious stimuli (* *p* < 0.05, ** *p* < 0.01, *** *p* < 0.001).

**Table 1 biomedicines-13-02241-t001:** Participants’ demographics and clinical characteristics.

	Stroke	Health	*t*	*p*
Sample size	30 (2 female)	35 (2 female)		
Age (years)	55.67 ± 11.31	54.89 ± 11.00	0.282	0.779
Age range (years)	27–77	27–77		
Education (years)	17.50 ± 4.62	15.94 ± 3.29	1.581	0.119
Age of stroke onset (years)	55.67 ± 11.31	–		
Stroke duration (days)	11.07 ± 8.13	–		
MRS score	0.29 ± 0.53	–		
NIHSS score	1.31 ± 1.47	–		
Room temperature (°C)	24.36 ± 1.82 (Exp1)	23.05 ± 1.52	3.152	0.002 *
	24.79 ± 1.75 (Exp2)	23.86 ± 1.51	2.274	0.026 *
Right foot temperature (°C)	35.75 ± 0.61 (Exp1)	35.91 ± 0.44	−1.191	0.238
	35.77 ± 0.67 (Exp2)	35.44 ± 1.05	1.503	0.138
Left foot temperature (°C)	35.73 ± 0.79 (Exp1)	35.90 ± 0.38	−1.002	0.312
	35.94 ± 0.45 (Exp2)	35.88 ± 0.40	0.609	0.545
PSQ score	4.35 ± 1.92	5.03 ± 1.56	−1.594	0.116
Pain threshold				
Heat (°C)	45.33 ± 2.77	44.80 ± 1.75	0.883	0.382
Pressure (N)	197.04 ± 43.59	178.30 ± 45.35	1.620	0.111
Stimulation intensity (Exp2)				
Heat (°C)	47.49 ± 1.83	46.87 ± 2.25	1.192	0.238
Pressure (N)	214.40 ± 68.30	184.33 ± 84.27	1.563	0.123

Note: MRS = modified. Rankin Scale; NIHSS = National Institute of Health Stroke Scale; PSQ = pain sensitivity scale; Exp1 = Experiment 1; Exp2 = Experiment 2; * *p* < 0.05.

**Table 2 biomedicines-13-02241-t002:** Comparisons of pain perception between the two groups (* *p* < 0.05).

Heat Intensity	M ± SD		*t*	*p*	Pressure Intensity	M ± SD		*t*	*p*
	Stroke	Health				Stroke	Health		
42 °C	1.07 ± 1.15	0.93 ± 0.88	0.55	0.583	100 N	2.14. ± 0.96	2.50 ± 1.44	–1.20	0.236
43 °C	1.11 ± 1.11	1.21 ± 1.01	−0.37	0.710	120 N	2.75 ± 1.21	3.48 ± 1.68	−1.97	0.054
44 °C	1.52 ± 1.20	1.37 ± 1.17	0.49	0.630	140 N	3.38 ± 1.48	4.16 ± 1.90	−1.82	0.074
45 °C	2.49 ± 1.84	2.79 ± 1.65	−0.70	0.488	160 N	3.70 ± 1.50	4.32 ± 1.74	−1.55	0.127
46 °C	3.17 ± 2.12	3.88 ± 2.17	−1.33	0.190	170 N	4.08 ± 1.62	5.03 ± 1.85	−2.18	0.033 *
47 °C	4.25 ± 2.58	4.91 ± 2.54	−1.04	0.303	180 N	4.37 ± 1.77	5.43 ± 1.92	−2.31	0.024 *
48 °C	4.92 ± 2.30	5.41 ± 2.08	−0.83	0.412	210 N	5.00 ± 1.99	6.10 ± 1.83	−2.34	0.023 *
49 °C	6.29 ± 2.43	6.80 ± 2.15	−0.83	0.410	220 N	5.45 ± 2.09	6.55 ± 1.92	−2.20	0.032 *
50 °C	7.33 ± 2.71	7.82 ± 2.27	−0.74	0.465	250 N	6.03 ± 2.32	7.04 ± 2.05	−1.87	0.066

## Data Availability

The raw data supporting the conclusions of this article will be made available by the authors on request.

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
