# Peer review of "Modalities Differentiation of Pain Perception Following Ischemic Stroke: Decreased Pressure Pain Perception"

_biomedicines, 2025, doi:10.3390/biomedicines13092241_

Round 1

Reviewer 1 Report

Comments and Suggestions for Authors

The authors study changes in pain perception in ischemic stroke patients using heat and pressure stimuli. Two experiments were conducted to assess pain, somatosensory thresholds, and to evaluate responses to both individual and simultaneous stimuli at sub- and supra-thresholds. The topic is timely, clinically relevant, and scientifically significant. The manuscript is well-written and informative.

However, in the opinion of this reviewer, some aspects of the ms could be improved to enhance clarity and scientific rigor.

(1) The title should explicitly state “ischemic stroke” rather than using the general term “stroke”.

(2) It would be a better idea to use keywords other than those already included in the title.

(3) Lines 48-51: the authors state, “(…) Multimodal somatosensory processing involves various types of nerve fibers [13]. Small fibers, such as C and Aδ, primarily respond to thermal, mechanical, and chemical stimuli [14], while large fibers, like Aβ, are associated with sensations of touch and pressure on the skin [15] (…)”. The authors should cite the original references for this information (e.g., DOI: 10.1016/j.cell.2009.09.028).

(4) In materials and methods section, the authors should clearly indicate the number of male and female participants in each experiment.

(5) There is no information about the statistical analyses used. This point should be included in the materials and methods section.

(6) The discussion is well-structured and coherent. However, if the authors believe it relevant, they might briefly discuss how these findings compare to or differ from what might be expected in cases of hemorrhagic stroke, since the current study only addresses ischemic stroke.

(7) One of the study’s limitations is the small sample size, particularly the low number of female patients. It would be helpful whether the authors address this limitation and its impact on the findings.

(8) Line 246: In the sentence “The current study investigated the difference in multimodal pains in patients”, the term “multimodal pains” is unclear. The authors should clarify what is meant by this term to avoid misinterpretation.

Reviewer 2 Report

Comments and Suggestions for Authors

Dear sir,

I have read your paper with interest, there are some methodological flaws so as to derive a valid scientific conclusion. 

I have following comments to offer.

1. All the patients were given stimulus (heat pain/pressure pain) to both the sides irrespective of affected site, however, in stroke patients the altered perception is assumed to be on the side of paresis or contralateral to the site of lesion, so giving pain stimulus on unaffected site will not give proper assessment of dysregulation.

2. In patients who were diabetic, were any of them having symptoms of peripheral neuropathy? As diabetes is known to be associated with neuropathy and is a well known risk factor for stroke and therefore will act as confounder, were these patients excluded from the study?

3. Instead of taking stimulus site on lower limb, it would have been better if upper limb was chosen as it's a well known fact that sensory perception is more fine in upper limb compared to lower limb. 

4. It would be more appropriate if the abnormal site was tested (for both heat and pressure pain) that too on upper limb and compared with normal site (not affected by stroke)of the same individuals well as with healthy control.

5. Altered pain perception in post stroke patient varies according to the site of lesion, and therefore, subgroup analysis based on lesion in central spinothalamic pathway compared to lesion outside this pathway should have been done for better understanding.

6. How many of these patient had residual deficit in form of spasticity or stiffness, as this will interfere with the stimulus perception and incorrect reporting of pain by the patient.

7. Pain perception of right sided noxious stimulus will go by left sided pathways, and vice-versa, so where is this integration of multimodal sensory processing expected and how it differs for healthy controls was not clearly mentioned.

8. About 20-30% patients as well as control could not tolerate higher temperatures, so it would have been better if same study was done on normal individuals as a pilot study to derive normative data, followed by performing this study using the normative data.

9. Correlation of pain perception with duration of stroke, across different sensory modalities should have been done.

10. How many of the participants (including healthy controls) had history of migraine, as migraineurs are known to have altered central pain perception and would act as confounder in this study. 

11. Clinical and radiological data of patients should be mentioned in results in one paragraph for better understanding of baseline status.

Reviewer 3 Report

Comments and Suggestions for Authors

Title: Modalities Differentiation of Pain Perception Following Stroke: Decreased Pressure Pain Perception

This study focuses on ischemic stroke patients and addresses the need to clarify somatosensory impairments and abnormal pain patterns, which are often governed by different neural pathways but have not been clearly presented in clinical settings. Research on stroke-induced changes in pain perception is necessary, especially studies that apply multiple sensory modalities—such as heat (activating C & Aδ fibers) and pressure (Aβ fibers)—to comprehensively assess the multidimensional nature of pain perception.
This approach adds important value to the field, and I believe with some revisions, the manuscript will be suitable for publication in Biomedicines.

Abstract

Methods:
The methodology section should clearly state the study design, participants, research period, and measurement tools. Currently, only the measurement tools for Experiments 1 and 2 are partially described, and this needs to be supplemented.
Please include the names of the instruments used.

Results:
It would be helpful to present the key findings of this study with specific numerical data so that readers can understand the results more intuitively.

2. Materials and Methods

  • Please add the manufacturing year of the measurement equipment.
    For example:
    - Medoc 9-cm² Contact Heat-Evoked Potential Stimulator (CHEPS; Medoc Ltd, Ramat Yishai, Israel) → Medoc 9-cm² Contact Heat-Evoked Potential Stimulator (CHEPS; Medoc Ltd, Ramat Yishai, Israel, [year])

  • Please provide information on the reliability and validity of the equipment, especially regarding its application in neurological patients such as those with stroke.

  • When the skin is exposed to repeated or prolonged heat stimuli, sensory receptors (particularly C-fibers and Aδ-fibers) may undergo desensitization, resulting in reduced or delayed pain perception to the same temperature stimulus.
    How was this thermal adaptation addressed in your study?
    Were there any participants who appeared to respond with artificially elevated pain thresholds due to adaptation?

  • You mentioned that each stimulus lasted for 5 seconds.
    Was there a rest period after each stimulus? If so, how long was it?

  • For heat stimulation, nine different temperature intensities were used.
    Were these presented in random order?

  • Considering the total of 45 stimuli, it seems like a large number.
    How did you control for possible adaptation due to cumulative stimulation?

  • The experimental design involves a progressive increase in temperature, which may lead to thermal adaptation of the skin.
    What measures were taken to mitigate or account for this effect?

  • The stimulus site was fixed (3 cm above the middle toe of the right foot).
    Given the possibility of local sensory adaptation due to repeated stimulation in one spot,
    how was this controlled for or compensated?

  • Overall, how did you address sensory adaptation and cumulative stimulation effects in your experimental design?

I would like to raise similar questions regarding the pressure stimulation protocol as well.

Round 2

Reviewer 1 Report

Comments and Suggestions for Authors

The authors have adequately addressed all my concerns.

Author Response

We sincerely thank the Reviewer for their time and for this positive feedback on our revised manuscript. We are very pleased that our responses and revisions have adequately addressed all of their previous concerns.

Reviewer 2 Report

Comments and Suggestions for Authors

Dear sir,

I have reviewed your paper, all the questions raised previously were not addressed. It has got methodological flaws to derive any valid scientific conclusions.

Author Response

We thank the Reviewer for their time and for providing further feedback on our manuscript.

We are, however, deeply concerned and somewhat perplexed by the comment that “all the questions raised previously were not addressed” and that the manuscript contains “methodological flaws.” In our previous revision, we provided a point-by-point response to all the specific comments raised by the reviewers and made extensive revisions to the manuscript, which we believed had satisfactorily addressed the concerns.

Regarding the general concern about “methodological flaws,” we have carefully re-examined our study design and statistical approaches and believe them to be sound and appropriate for testing our hypotheses. However, without more specific details from the Reviewer on what these purported flaws entail, we find it exceedingly difficult to provide a targeted response or to undertake further constructive revisions.

We respectfully request that the Reviewer could kindly provide more precise guidance regarding:

Which specific previous points they feel remain unaddressed?

The nature of the alleged methodological flaws, so that we might either address them thoroughly or engage in a constructive scientific discussion.

We are fully committed to improving the quality of our work and sincerely wish to address any legitimate scientific concerns. We are happy to undertake further revisions if provided with specific, actionable feedback.

We kindly leave it to the Editor's discretion to judge whether our previous revisions are adequate or if, in light of this vague but severe criticism, further specific details from the Reviewer are necessary to proceed.

Reviewer 3 Report

Comments and Suggestions for Authors

Dear Authors,

This manuscript presents an investigation into ischemic stroke patients, emphasizing the importance of clarifying somatosensory impairments and abnormal pain patterns—phenomena often mediated by distinct neural pathways but insufficiently characterized in clinical practice. The focus on stroke-related alterations in pain perception is both timely and relevant, particularly through the application of multiple sensory modalities, such as heat (activating C and Aδ fibers) and pressure (stimulating Aβ fibers), to capture the multidimensional aspects of pain perception.

I appreciate your thorough and thoughtful revisions in response to the questions raised during the review process. In my assessment, the manuscript meets the scientific and methodological standards required for publication in Biomedicines.
Thanks.

Author Response

We sincerely thank the Reviewer for their exceptionally positive and insightful comments on our manuscript. We are greatly encouraged by their assessment that our work is "timely and relevant" and that it meets the publication standards of Biomedicines. We are also very grateful that they acknowledged our efforts in addressing the previous points raised during review. Our appreciation again for their time and for this valuable feedback.